# C-X-C Motif Chemokine Ligand 9 and Its CXCR3 Receptor Are the Salt and Pepper for T Cells Trafficking in a Mouse Model of Gaucher Disease

**DOI:** 10.3390/ijms222312712

**Published:** 2021-11-24

**Authors:** Albert Frank Magnusen, Reena Rani, Mary Ashley McKay, Shelby Loraine Hatton, Tsitsi Carol Nyamajenjere, Daniel Nii Aryee Magnusen, Jörg Köhl, Gregory Alex Grabowski, Manoj Kumar Pandey

**Affiliations:** 1Division of Human Genetics, Cincinnati Children’s Hospital Medical Center, 3333 Burnet Avenue, Cincinnati, OH 45229, USA; albert.magnusen@cchmc.org (A.F.M.); Mary_McKay@rush.edu (M.A.M.); shelby.hatton@cchmc.org (S.L.H.); nyamajtc@mail.uc.edu (T.C.N.); daniel.magnusen@sinclair.edu (D.N.A.M.); 2Division of Immunobiology, Cincinnati Children’s Hospital Medical Center, 3333 Burnet Avenue, Cincinnati, OH 45229, USA; reenacchmc@gmail.com; 3Institute for Systemic Inflammation Research, University of Lübeck, 23562 Lübeck, Germany; joerg.koehl@uksh.de; 4Department of Pediatrics and Division of Immunobiology, Cincinnati Children’s Hospital Medical Center, College of Medicine, University of Cincinnati, 3333 Burnet Avenue, Cincinnati, OH 45229, USA; 5Department of Molecular Genetics, Biochemistry and Microbiology, Division of Human Genetics, Cincinnati Children’s Hospital Medical Center, College of Medicine, University of Cincinnati, 3333 Burnet Avenue, Cincinnati, OH 45229, USA; grabgo317@comcast.net; 6Department of Pediatrics, Division of Human Genetics, Cincinnati Children’s Hospital Medical Center, College of Medicine, University of Cincinnati, 3333 Burnet Avenue, Cincinnati, OH 45229, USA

**Keywords:** lysosomal storage disease, chemokine, chemokine receptor, inflammation

## Abstract

Gaucher disease is a lysosomal storage disease, which happens due to mutations in *GBA1*/*Gba1* that encodes the enzyme termed as lysosomal acid β-glucosidase. The major function of this enzyme is to catalyze glucosylceramide (GC) into glucose and ceramide. The deficiency of this enzyme and resultant abnormal accumulation of GC cause altered function of several of the innate and adaptive immune cells. For example, augmented infiltration of T cells contributes to the increased production of pro-inflammatory cytokines, (e.g., IFNγ, TNFα, IL6, IL12p40, IL12p70, IL23, and IL17A/F). This leads to tissue damage in a genetic mouse model (*Gba1*^9V/−^) of Gaucher disease. The cellular mechanism(s) by which increased tissue infiltration of T cells occurs in this disease is not fully understood. Here, we delineate role of the CXCR3 receptor and its exogenous C-X-C motif chemokine ligand 9 (CXCL9) in induction of increased tissue recruitment of CD4^+^ T and CD8^+^ T cells in Gaucher disease. Intracellular FACS staining of macrophages (Mϕs) and dendritic cells (DCs) from *Gba1*^9V/−^ mice showed elevated production of CXCL9. Purified CD4^+^ T cells and the CD8^+^ T cells from *Gba1*^9V/−^ mice showed increased expression of CXCR3. Ex vivo and in vivo chemotaxis experiments showed CXCL9 involvement in the recruitment of *Gba1*^9V/−^ T cells. Furthermore, antibody blockade of the CXCL9 receptor (CXCR3) on T cells caused marked reduction in CXCL9- mediated chemotaxis of T cells in *Gba1*^9V/−^ mice. These data implicate abnormalities of the CXCL9-CXCR3 axis leading to enhanced tissue recruitment of T cells in Gaucher disease. Such results provide a rationale for blockade of the CXCL9/CXCR3 axis as potential new therapeutic targets for the treatment of inflammation in Gaucher disease.

## 1. Introduction

Gaucher disease (GD) is a lysosomal storage disorder with a worldwide incidence of approximately 1/40,000 to 1/100,000 [1,2]. GD is caused by *GBA1* mutations that lead to decreased activity of lysosomal acid β-glucosidase (D-glucosyl-N-acylsphingosine glucohydrolase (EC 4.2.1.25; GCase) and the resultant abnormal tissue accumulation of glucosylceramide (GC) [3,4]. Macrophage (Mϕ) lineage cells are prominent disease effectors due to their massive accumulation of GC, i.e., Gaucher cells. This leads to their secretion of numerous cytokines and chemokines that influence other innate and adaptive immune cells [5,6,7,8]. The resultant tissue manifestations of GD lead to the clinical signs of anemia, thrombocytopenia, hypergammaglobulinemia, splenomegaly, hepatomegaly, bone, and brain defects) [1,6,9,10,11,12,13,14]. Many of these signs are recapitulated in a genetic GD-mouse model (D409 V/null; 9 V/null; *Gba1*^9V/−^) including, tissue accumulation of Gaucher cells and GC in lung, liver, and spleen as well as infiltration of T cells [4,5,13,15,16,17,18,19,20,21,22].

T lymphocytes are the major effector cells in cellular immunity and produce cytokines in response to variety of antigens, which leads to inflammation in several diseases [23,24,25,26,27]. Two major groups T cells are termed CD4^+^ T-helper and CD8^+^ T cytotoxic cells [28]. CD4^+^ T helper cells showed significant heterogeneity of their cytokine expression profiles that lead to the discoveries of interferon gamma (IFNγ) producing T helper 1 (Th1), interleukin (IL) 4 producing Th2, and IL17 producing Th17 cell subsets [26,29,30]. This cytokine heterogeneity specifies the interaction of T cells with other immune cells and thereby their function in host defense and inflammation [25,31,32,33,34,35,36]. CD8^+^ T cells are important for inducing autoimmune and the anti-cancer and anti-viral responses [37,38,39,40]. T cell defects, T cell lymphomas, and increased incidence of CD3^+^, CD4^+^, CD8^+^, CD3^+^HLA-DR^+^, CD4^+^HLA-DR^+^, and CD8^+^HLA-DR^+^ subset of activated T cells have been observed in lung and peripheral blood of patients with GD [41,42,43,44,45,46]. Significantly elevated levels of CD4^+^ T cells and modest changes in CD8^+^ T cells were present in liver, lung, spleen, and thymus of *Gba1*^9V/−^ mice [5,6,47]. The CD3/CD28-mediated GC-dependent activation of liver-, lung-, and spleen-derived T cells in co-culture of DC and CD4^+^ T cells have shown increased production of several of the pro-inflammatory cytokines, i.e., IFNγ, tumor necrosis factor alfa (TNFα), IL6, IL12p40, IL12p70, IL23, and IL17A/F. This pro-inflammatory environment leads to the tissue damage in *Gba1*^9V/−^ GD mouse model. The genetic deficiency or pharmaceutical blockade of complement 5a (C5a) receptor 1 (C5aR1) resulted in reduction of activated subsets of CD4^+^ T cells as well as the decreased generation of pro-inflammatory cytokines in co-cultured cells, (e.g., DC and CD4^+^ T cells) from *Gba1*^9V/−^ mice [16]. However, the exact cellular mechanism(s) that causes enhanced tissue recruitment of T cells in *Gba1*^9V/−^ mice is still unclear.

C-X-C motif ligand 9 (CXCL9), CXCL10, and CXCL11 belong to the CXC subfamily of chemokines. These are induced by IFNγ and are crucial for recruitment of T cells and other immune cell phenotypes, (e.g., NK cells) to the sites of inflammation, due to their binding to chemokine receptor CXCR3 [48,49,50,51,52,53,54,55,56,57,58,59,60]. CXCR3 is the member of G protein coupled receptor family and is expressed on different subsets of T cells [61,62,63,64]. IFNγ drives increased expression of CXCR3 and its ligands, (e.g., CXCL9–11) that are increased in several inflammatory diseases [64,65,66,67,68,69,70]. Elevated levels of IFNγ and CXCL9–11 as well as increased numbers of tissue T cells were found in GD mouse models and human patients with GD [5,6,15,41,42,43,44,45,47,71]. However, the cellular mechanism(s) that causes increased tissue recruitment of T cells in *Gba1*^9V/−^ mice is unclear. Here, analyses of CXCL9-mediated ex vivo and in vivo T cells chemotaxis in the presence or absence of mouse anti-CXCR3 antibodies identified a role of the CXCL9-CXCR3 axis in excess trafficking of T cells into tissues affected by GD.

## 2. Materials and Methods

### 2.1. Materials

The following reagents were from BD Biosciences (San Jose, CA, USA) or eBiosciences (San Diego, CA, USA): Monoclonal antibodies to CD11b-FITC (M1/70), F4/80-PerCP5.5, CD11c-APC, CD3-pacific blue, CD4-FITC, CD8-APC, CXCL9-PE, CXCR3/CD183-PE and their corresponding isotype antibodies (Rat IgG2a-FITC, Rat IgG2a-PerCP5.5, Armenian hamster IgG-APC, Rat IgG2a-pacific blue, Rat IgG2a-FITC, Rat IgG2a APC, Hamster IgG-PE). Recombinant murine CXCL9 from Pepro Tech (Cranbury, NJ, USA) and purified mouse anti-CXCR3 antibodies (catalog number-155902, clone-S18001A, and lot number-B265189) was from Biolegend (San Diego, CA, USA). Liberace Cl was from Roche (Indianapolis, IN, USA). Bovine serum albumin (BSA), Gey’s balanced salt solution (GBSS), and DNase were from Sigma (St. Louis, MO, USA). Anti-CD11b, CD11c, CD4, and CD8 microbeads were from Miltenyi Biotec (Auburn, CA, USA). Diff-Quik stain set was from Dade Behring, Inc. (Newark, NJ, USA). Polycarbonate membranes, cell scraper, and Boyden chemotaxis chamber were from Neuro Probe, Inc. (Gaithersburg, MD, USA). ELISA kit for the detection of mouse CXCL9 was from R&D System (Minneapolis, MN, USA). LSRII flow cytometer from BD Biosciences (San Jose, CA, USA), FCS Express software from De Novo Software (Los Angeles, CA, USA).

### 2.2. Mice

The D409 V/null mice (9 V/null; *Gba1*^9V/−^) and WT control were of the mixed background FVB/C57BL/6J/129SvEvBrd (50:25:25) [4] and were 12 weeks of age. The new nomenclature for D409V includes the 39 amino acid leader sequence and would then be Asp448Val or p.D448V. Mice were maintained under pathogen-free conditions. All mice were housed under pathogen-free conditions in the barrier animal facility according to IACUC-approved protocol (IACUC2020-0052) at Cincinnati Children’s Hospital Research Foundation (CCHRF).

### 2.3. Cell Preparation

Lung, spleen, blood, and peritoneal lavage from WT and *Gba1*^9V/−^ mice were removed aseptically. Single cell suspensions prepared from lung were obtained from minced pieces that were treated with Liberase Cl (0.5 mg/mL) and DNase (0.5 mg/mL) in RPMI (45 min, 37 °C) and spleen by direct grinding. Blood mononuclear cells were obtained after red blood cell (RBC) lysis (155 mM NH_4_Cl, 10 mM NaHCO_3_, 0.1 mM EDTA). Single cell suspensions prepared from lung, spleen, and the peritoneal lavage were filtered through a 70-micron cell strainer followed by RBC lysis, passage through a strainer, and pelleted by centrifugation at 350 g. Viable cells were counted using a Neubauer chamber and trypan blue exclusion. Mϕs, DCs, CD4^+^ T lymphocytes, and CD8^+^ T lymphocytes were purified from single cell suspensions of lung and spleen using CD11c, CD11b, CD4 (L3T4), and CD8a (Ly2) microbeads according to the manufacturer’s protocol.

### 2.4. Flow Cytometry

FACS staining was performed for characterization of immunological cell types in lung, spleen, blood, and peritoneal lavage and the chemotactic cells obtained after their migration. These cells were washed with PBS containing 1% BSA. After incubation for 15 min at 4 °C with the blocking antibody 2.4G2 (anti-FcγRII and III), all cells were stained at 4 °C for 45 min with the appropriate labeled antibodies for different cell types, i.e., anti-mouse CD11b and anti-mouse F4/80 antibodies for Mϕs, anti-mouse CD11b, anti-mouse CD11c antibodies for DCs, anti-mouse CD3, CD4, and CD8, antibodies for T cells, and anti-mouse B220 antibodies for B cells. In separate batches, the cells were stained with the respective isotypes. Flow cytometric analyses were performed, where Mϕs were gated first by their typical FSC/SSC pattern based on monocyte gated cells and their F4/80 positivity and double stained for F4/80 and CD11b. For DCs, monocyte gated cells from FSC/SSC pattern were gated for CD11c positivity and double stained for CD11c and CD11b. Purified Mϕs and DCs were used to perform intracellular cytokine staining for CXCL9 and its isotype, (e.g., Armenian Hamster IgG). Flow cytometric analyses of T lymphocytes were generated after gating lymphocytes from forward and side scatter and then identifying the CD3^+^, B220^−^CD3^+^CD4^+^, and B220^−^CD3^+^CD8^+^ T lymphocytes. Mononuclear cells prepared from blood as well as purified CD4^+^ T and CD8^+^ T cells were used to perform surface staining for CXCR3 and its isotype, (e.g., Armenian Hamster IgG). In an addition experiment, purified CD4^+^ T cells and CD8^+^ T cells were used for chemotaxis assays. Flow cytometric analyses were performed on a LSR II, and FCS Express software.

### 2.5. T Cell Chemotaxis

CD4^+^ T cells prepared from spleen of WT and *Gba1*^9V/−^ mice were suspended in chemotaxis medium (GBSS containing 2% BSA) at a density of 5 × 10^6^ cells/mL. The different concentration of CXCL9, (e.g., 0, 2, 4, 8, 16, and 32 nM) in chemotaxis medium, placed in the bottom wells of a micro-Boyden chambers and overlaid with a 3 µm polycarbonate membrane. Then, 50 µL of the cells were placed in the top wells and incubated for 45 min at 37 °C. Subsequently, the membranes were removed and the cells on the bottom side of the membrane were stained with Diff-Quick. The numbers of migrated cells in five high-power fields were counted and the number of cells per mm^2^ was calculated by computer assisted light microscopy. Results are expressed as the mean value of triplicate samples.

### 2.6. Ex Vivo Blocking of CXCR3 and T Cells Chemotaxis

To examine whether ex vivo blocking of CXCR3 using mouse anti-CXCR3 antibodies can reduce CXCL9 mediated chemotaxis of T cell subsets in *Gba1*^9V/−^ GD model, spleen-derived CD4^+^ T cells and CD8^+^ T cells (5 × 10^6^ cells/mL) prepared from WT and *Gba1*^9V/−^mice were treated in the presence and absence of antibodies to mouse CXCR3 (10 μg/mL) at 4 °C for 30 min. These cells were applied to subsequent top wells of Boyden chemotaxis chamber and chemotaxis was performed towards CXCL9 (16 nM) at 37 °C and 5% CO_2_ for 45 min. The membrane was removed, and cells were scraped off using a vertical glass slide on the top of 50 mL falcon tube. These cells were stained with antibodies to specific cell phenotypes as discussed above.

### 2.7. In Vivo Blocking of CXCR3 and T Cells Chemotaxis

To examine whether in vivo blocking of CXCR3 alters CXCL9- mediated increased tissue recruitment of T cells in Gaucher disease, WT (*n* = 5) and *Gba1*^9V/−^ mice (*n* = 5) were injected intraperitoneally (IP) with CXCL9 (200 nM:100 μL) and their vehicle (PBS). In some experiments, mice were pretreated intravenously (IV) with mouse anti-CXCR3 antibodies (1.0 mg/kg body weight) prior to IP injections of CXCL9 or vehicle (PBS). After 6 h, mice were killed, and the peritoneal cavity was lavage with 10 mL of PBS. Peritoneal cells were washed once with PBS, and 10^5^ cells in 200 μL of PBS were used for performing FACS staining with antibodies to mouse CD3, CD4, and CD8s.

### 2.8. Determination of CXCL9 Production

Mϕs and DCs purified from lung of the strain-matched *Gba1*^9V/−^ and WT mice were cultured (10^6^ cells/200 μL of complete RPMI media) for 48 h. CXCL9 concentrations were determined in the cell supernatants by commercial ELISA kits according to the manufacturer’s instructions.

### 2.9. Statistical Analyses

Statistical significance was assessed by either one-tailed Students t-test (two groups) or analysis of variance (ANOVA) for multiple groups to determine significance performed using Prism Graph Pad™. Values shown in one asterisk (*, *p* < 0.05); two asterisks (**, *p* < 0.01); three asterisks (***, *p* < 0.001), and four asterisks (****, *p* < 0.0001) were considered statistically significant.

## 3. Results

### 3.1. Gba1^9V/−^ Mice Immune Phagocytes Show Increased Levels of CXCL9 Chemokines

Mϕs and DCs purified from lungs of WT and *Gba1*^9V/−^ mice were used to assay CXCL9 chemokine levels. Compared to WT mouse Mϕs, *Gba1*^9V/−^ mouse Mϕs had significantly increased amounts of CXCL9 (Figure 1a–e; *p* < 0.0001). Similarly, as compared to WT mice, *Gba1*^9V/−^ mouse DCs showed significantly increased amounts of CXCL9 (Figure 1f–j; *p* < 0.0001).

### 3.2. Identification of CXCR3 Positive T Cells in Gba1^9V/−^ Mice

The single cell suspensions prepared from blood of WT and *Gba1*^9V^^/-^ mice were analyzed for CXCR3^+^CD3^+^ T cells. Compared to WT mouse samples, those from *Gba1*^9V/−^ mice CD3^+^ T cells had elevated amounts of CXCR3 (Appendix A; *p* < 0.0001). An additional experiment CD4^+^ T cells and CD8^+^ T cells purified from lung of WT and *Gba1*^9V/−^ mice were analyzed for CXCR3. Compared to WT mice, *Gba1*^9V/−^ mouse CD4^+^ T cells had elevated CXCR3 (Figure 2a–d; *p* < 0.0001). In addition, as compared to WT, *Gba1*^9V/−^ mouse CD8^+^ T cells had elevated CXCR3 (Figure 2e–h; *p* < 0.0001).

### 3.3. Effect of CXCL9 in Ex Vivo Chemotaxis of T Cells in Gba1^9V/−^ Mice

*Gba1*^9V/−^ mice immune phagocytes, (e.g., Mϕs and DCs) showed increased amounts of CXCL9 and their receptor CXCR3 on T cell subsets when compared to WT. These data suggested a potential role of the CXCL9—CXCR3 pathway for increased numbers of T cells in *Gba1*^9V/−^ mouse tissues. To confirm this, several concentrations of CXCL9 (0, 2, 4, 8, and 16 nM) were used to generate dose response curves for ex vivo chemotaxis of WT and Gba1^9V/−^ mouse spleen-derived CD4^+^ T cells. CXCL9 caused dose-depended increase in chemotaxis of CD4^+^ T cells in WT and *Gba1*^9V/−^ mice; compared to WT, such effects were more pronounced in *Gba1*^9V/−^ mice (Figure 3a–c; *p* < 0.01; *p* < 0.0001).

### 3.4. Pharmaceutical Targeting of CXCR3 Leads to the Reduction of CXCL9 Mediated Ex Vivo T Cells Chemotaxis in Gba1^9V/−^ Mice

Pharmaceutical blocking of CXCR3 confirmed the altered CXCL9-mediated ex vivo T cell chemotaxis in *Gba1*^9V/−^ mice. Mouse anti-CXCR3 antibodies or vehicle (PBS) treated WT and *Gba1*^9V/−^ mouse spleen-derived CD4^+^ T cells were used for assessing their chemotaxis towards CXCL9 (Figure 4a–d). Similarly, mouse anti-CXCR3 antibodies or vehicle (PBS) treated WT and *Gba1*^9V/−^ mouse spleen-derived CD8^+^ T cells were used for assessing their chemotaxis towards CXCL9 (Figure 5a–d).

In additional experiments, anti-CXCR3 antibodies and vehicle (PBS) were used to treat WT and *Gba1*^9V/−^ mice spleen-derived CD4^+^ T and CD8^+^ T cells. These CD4^+^ T cells.

(Appendix A) and CD8^+^ T cells (Appendix A) were used for chemotaxis quantification towards the corresponding chemotaxis buffer, i.e., 2% BSA-GBSS.

Cells were separated and analyzed by flow cytometry. Compared to WT, analyses of cells from *Gba1*^9V/−^ mice that migrated towards CXCL9 showed increased percentages of CD4^+^CD11b^−^ (Figure 4a,c,e; *p* < 0.0001) and CD8^+^CD11b^−^ T cells (Figure 5 a,c,e; *p* < 0.0001). As compared to vehicle treated *Gba1*^9V/−^ cells, mouse anti-CXCR3 antibodies treated *Gba1*^9V/−^ cells showed reduction in CXCL9 mediated increased chemotaxis of CD4^+^CD11b^−^ (Figure 4c–e; *p* < 0.0001) and CD8^+^CD11b^−^ T cells (Figure 5c–e; *p* < 0.0001). However, these differences were not significant when compared to vehicle or mouse anti-CXCR3 antibodies treated WT CD4^+^CD11b^−^ (Figure 4a,b,e; ns) or WT CD8^+^CD11b^−^ T cells (Figure 5a,b,e; ns). Furthermore, vehicle or mouse anti-CXCR3 antibodies treated WT and *Gba1*^9V/−^ mouse CD4^+^CD11b^−^T (Appendix A; ns) or CD8^+^CD11b^−^T (Appendix A; ns) cells migration towards chemotaxis buffer did not differ.

### 3.5. Pharmaceutical Targeting of CXCR3 Causes the Reduction of CXCL9 Mediated In Vivo T Cells Chemotaxis in Gba1^9V/−^ Mice

To confirm if in vivo administration of mouse anti-CXCR3 antibodies decrease the CXCL9-mediated chemotaxis of T cell subsets in GD, WT and *Gba1*^9V/−^ mice were treated with CXCL9 and its vehicle (PBS) in the presence and absence of mouse anti-CXCR3 antibodies The peritoneal cells were analysed for total cell infiltrates as well as the CD3^+^CD4^+^ T cells and CD3^+^CD8^+^ T cells (see Methods). Compared to vehicle (PBS) or mouse anti-CXCR3 antibodies, administered CXCL9 to WT mice showed increased peritoneal cell recruitment (Appendix A; *p* < 0.01). Compared to administered CXCL9, mouse anti-CXCR3 antibodies given prior to CXCL9 injection abrogated the CXCL9 mediated increased recruitment of peritoneal cells in WT mice (Appendix A; *p* < 0.01). As compared to vehicle or mouse anti-CXCR3 antibodies administration, CXCL9 injected *Gba1*^9V/−^ mice showed more pronounced peritoneal cell infiltrates (Appendix A; *p* < 0.0001). Compared to administered CXCL9, mouse anti-CXCR3 antibodies given prior to CXCL9 injection caused marked reductions in the increased recruitment of peritoneal cells in *Gba1*^9V/−^ mice (Appendix A; *p* < 0.0001). These findings were also obtained for CD4^+^ T cells (Figure 6e–i; *p* < 0.0001, *p* < 0.001). In WT mice, the CD4^+^ T cell differences were not significant when compared with vehicle, mouse anti-CXCR3 antibodies, CXCL9 and mouse anti-CXCR3 antibodies administered prior to CXCL9 (Figure 6a–e; ns).

As compared to PBS or mouse anti-CXCR3 antibodies treated mice, CXCL9 injected *Gba1*^9V/−^ mice showed elevated recruitment of peritoneal CD8^+^ T cells (Figure 7e–h; *p* < 0.0001), In comparison, mouse anti-CXCR3 antibodies administration prior to giving CXCL9 abrogated the increased recruitment of peritoneal CD8^+^ T cells in *Gba1*^9V/−^ mice (Figure 7e,h,i; *p* < 0.05). The effects of these treatments on WT CD8^+^-cells were not significant (Figure 7a–e; ns).

## 4. Discussion

Here, Mϕs and DCs have been recognized as the sources of local increases of CXCL9 in *Gba1*^9V/−^ mice. Furthermore, CD4^+^ T cells and the CD8^+^ T cells from *Gba1*^9V/−^ mice had increased amounts of CXCR3. Although not explicitly tested, these findings implicate mutant *Gba1* and the resultant excess tissue accumulation of GC in increasing the production/expression of CXCL9 in GD. Such increased CXCL9 directed the chemotaxis of CXCR3 expressing T cell subsets in the *Gba1*^9V/−^mouse model of GD. This is supported by the lung-derived *Gba1*^9V/−^ Mϕs and DCs having increased amounts of CXCL9 and the T cell subsets with increased amounts of CXCR3.

In addition, these findings highlight the importance of the CXCL9-CXCR3 axis in the induction of ex vivo and in vivo chemotaxis of CD4^+^ and CD8^+^ T cells in the *Gba1*^9V/−^mouse.

CXCL9 chemokines attracts CXCR3^+^CD4^+^ and CD8^+^ effector T cells to sites of inflammation and direct their polarization into highly potent effector T cells that lead to the tissue enlargement in several diseases [64,67,72,73,74,75,76,77,78,79,80,81,82,83,84]. Elevated levels of such T cell subsets and their interaction with antigen presenting cells (e.g., DCs and/or Mϕs) are a and, potentially, the major effector contributing to massive increases of pro-inflammatory cytokines and tissue destruction in GD [5,6,15,16,85]. However, the cellular mechanism(s) underlying increased infiltration of T cell subsets in GD are not clearly understood. As compared to CXCL10 and CXCL11, massive increases of CXCL9 in *Gba1*^9V/−^ [10] provided the impetus for exploring the role of CXCL9-induced T cell trafficking in GD.

In certain tissues, e.g., lung, intestine, and tumor, cause downregulation of chemokine receptor expression once the infiltrating cells reside in the tissues and are exposed to high concentrations of ligands and/or interact with abnormal local production of pro-inflammatory cytokines [86,87,88,89]. To avoid this limitation, the current study used spleen derived T cells for testing CXCL9-mediated ex-vivo chemotaxis in the *Gba1*^9V/−^ mouse. This study identified a direct role of the CXCL9-CXCR3 axis in aiming the excess tissue recruitment of T cells in the *Gba1*^9V/−^ mouse. CXCR3 is an attractive therapeutic target for treating T cell-mediated inflammatory diseases [83,84,90,91,92,93,94,95,96]. CXCL9-mediated ex vivo and in vivo chemotaxis of T cell subsets (i.e., CD4^+^ T and CD8^+^ T cells) in the presence or absence of mouse anti-CXCR3 antibodies showed that targeting CXCR3 caused marked reduction in CXCL9-mediated enhanced tissue recruitment of T cell subsets in GD.

The exact mechanism(s) by which immune phagocytes, (e.g., Mϕs and DCs) and/or T cells lead to increased levels of CXCL9/CXCR3 in *Gba1*^9V/−^ mice remain to be fully elucidated. IFNγ and its downstream signaling is needed for driving CXCL9-CXCR3-mediated tissue inflammation in several diseases [64,65,66,67,68,69,70].

Excess tissue amounts of GC, IFNγ, and their downstream effects have been reported in mouse models and human patients with GD [5,6,13,15,16,17,41,42,43,44,45,47,71]. Additionally, genetic deficiency of C5aR1 resulted in decreased tissue levels of GC, IFNγ, CXCL9–11, and reductions in tissue recruitment of T cell subsets in mouse models of GD [16]. These findings implicate a mechanistic link between GC/C5a-C5aR1-IFNγ pathways for activation of CXCL9/CXCR3 in GD, which require further mechanistic elucidation.

The clinical features of human GD that happens due to *GBA1* defect includes anemia, thrombocytopenia, hypergammaglobulinemia, splenomegaly, hepatomegaly, bone and brain defects) [1,9,10,11,12,14]. *Gba1*^9V/−^ mouse model of GD recapitulated many of these signs including, tissue accumulation of Gaucher cells and GC in lung, liver, and spleen as well as infiltration of T cells [4,5,13,15,16,17,18,19,20,21,22]. However, there is still some limitations of the study as the *Gba1*^9V/−^ mouse model of GD does not recapitulate exactly the human disease and other indicated disease complications.

Mice express a single isoform of CXCR3 that exclusively binds to CXCL9, CXCL10, and CXCL11. CXCR3a, b and alt isoforms exist in humans [97]. Human CXCR3a is equivalent to mouse CXCR3 and binds CXCL9, CXCL10, and CXCL11. Human CXCR3b binds to CXCL9, CXCL10, CXCL11 as well as an additional ligand CXCL4. Human CXCR3alt binds specifically to CXCL11 [97]. The translational potential of this research could be challenging as in contrast to murine CXCL9 and CXCL11, human CXCL9–11 are inactivated rapidly in the presence of physiological concentrations of dipeptidyl peptidase IV/CD26 [98,99,100]. Despite of these complexities, the current study invites investigation into the different isoforms of CXCR3 and their ligands, i.e., CXCL9–11, as well as their up and/or downstream signaling that enhance T cell trafficking in GD. These findings open new areas of research that may identify CXCR3 and several of their ligands as interesting drug targets for modulation of immune cell function that fuel tissue inflammation in GD and other lysosomal storage sicknesses.

## Figures and Tables

**Figure 1 ijms-22-12712-f001:**
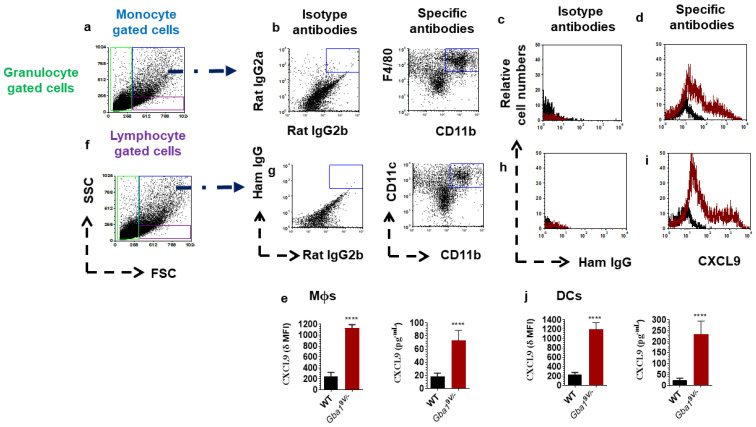
Immune phagocytes that cause increased amounts of CXCL9 in *Gba1*^9V/−^ mouse tissues. CXCL9 amounts in monocyte gated F4/80^hi^CD11b^+^ Mϕs (**a**–**e**) and CD11c^hi^CD11b^+^ DCs (**f**–**j**) from lung of strain-matched *Gba1*^9V/−^ and WT mice (*n* = 5/group). Delta Mean Fluorescence Intensity (δ MFI): CXCL9 MFI—isotype MFI. In the histograms of isotypes (**c**–**h**), specific antibodies (**d**–**i**), and the bar diagrams, the black lines/columns correspond to WT and the maroon lines/columns to *Gba1*^9V/−^ cell. Values in d-h are the means ± SD. and asterisks show significant differences between WT and *Gba1*^9V/−^ mice (**** *p* < 0.0001). Three independent experiments were conducted, and groups were compared using student’s *t*-tests.

**Figure 2 ijms-22-12712-f002:**
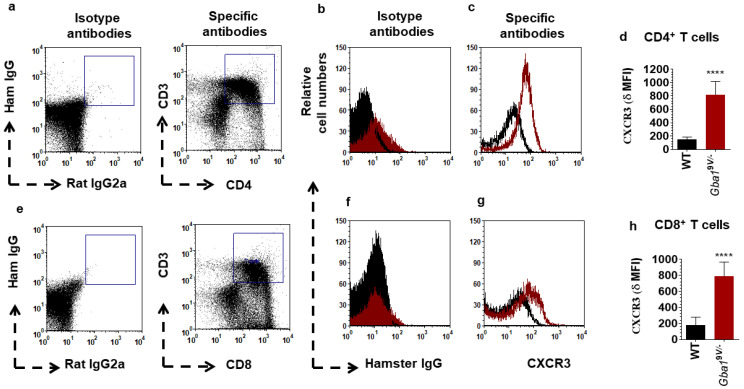
CXCR3 surface expression in pulmonary T cell subsets from *Gba1*^9V/−^ mice. CXCR3 expression in FACS-sorted CD3^+^CD4^+^ T cells (**a**–**d**) and CD3^+^CD8^+^ T cells (**e**–**h**) from lung of strain-matched *Gba1*^9V/−^ and WT mice (*n* = 5/group). δ MFI: CXCR3 MFI—isotype MFI. In the dot plots of isotypes and specific antibodies (**a**,**e**), in the histograms of isotypes (**b**,**f**), specific antibodies (**c**,**g**), and the bar diagrams (**d**,**h**), the black lines/columns correspond to WT and the maroon lines/columns to *Gba1*^9V/−^ mice. Values in d and h are the means ± SD. and asterisks show significant differences between WT and *Gba1*^9V/−^ mice (**** *p* < 0.0001). Three independent experiments were conducted, and groups were compared using student’s *t*-tests.

**Figure 3 ijms-22-12712-f003:**
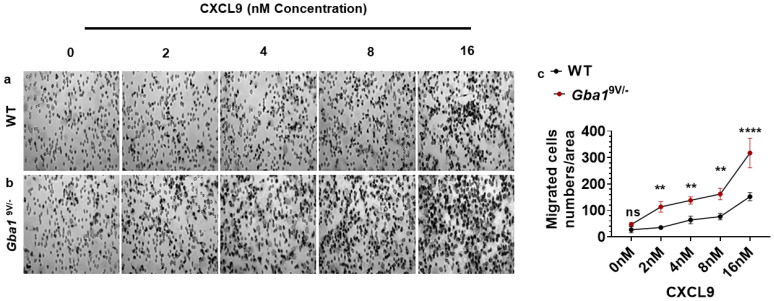
CXCL9 drives dose-dependent increased chemotaxis of T cell subsets in *Gba1*^9V/−^ mice. Spleen-derived CD4^+^ T cells from WT and *Gba1*^9V/−^ mice (*n* = 5/group) were allowed to migrate towards different concentrations of CXCL9 (0, 2, 4, 8, and 16 nM) at 37 °C and 5% CO_2_ for 45 min. The migration membrane was removed and stained with Diff-Quick and cells were counted under the light microscope. Diff quick- images and the corresponding bar diagrams represent the CXCL9 induced T cells migration in WT (**a**–**c**) and *Gba1*^9V/−^ mice (**b**,**c**). WT (black curve), *Gba1*^9V/−^ (maroon curve) and values shown are the mean ± SD. and group comparison were performed with ANOVA (ns, not significant; **, *p* < 0.01; ****, *p* < 0.0001).

**Figure 4 ijms-22-12712-f004:**
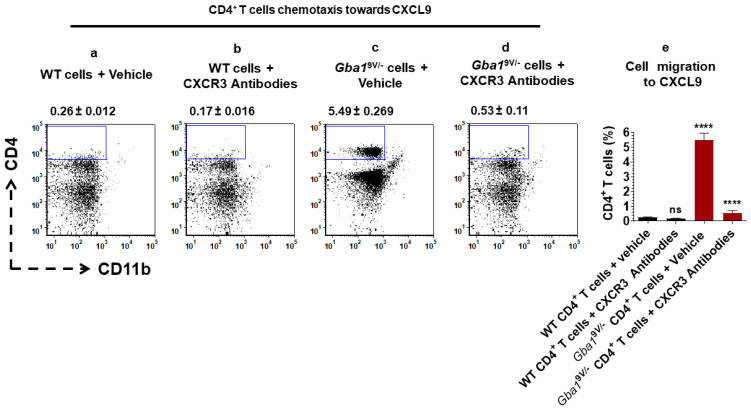
CXCR3 targeting alters CXCL9-mediated CD4^+^ T cells chemotaxis in *Gba1*^9V/−^ mice. CD4^+^ T cells purified from spleen of WT and *Gba1*^9V/−^ mice (*n* = 5/group) were allowed to migrate towards CXCL9 (16 nM) in the presence and absence of mouse anti-CXCR3 antibodies (10 μg/mL) at 37 °C and 5% CO_2_ for 45 min. Cells that had migrated through the filter and had attached to the lower side of the filter were collected and analyzed by FACS. Percentage of CD4^+^CD11b^−^ T cells are shown from the (**a**) vehicle (PBS) treated WT cells and their migration to CXCL9, (**b**) mouse anti-CXCR3 antibodies treated WT cells and their migration to CXCL9, (**c**) PBS treated *Gba1*^9V/−^ cells and their migration to CXCL9, and (**d**) mouse anti-CXCR3 antibodies treated *Gba1*^9V/−^ cells and their migration to CXCL9. (**e**) WT (black columns), *Gba1*^9V/−^ (maroon columns) and the values shown in the bar diagram are the mean ± SD. and group comparison were performed with ANOVA. Three independent experiments were conducted (**** *p* < 0.0001).

**Figure 5 ijms-22-12712-f005:**
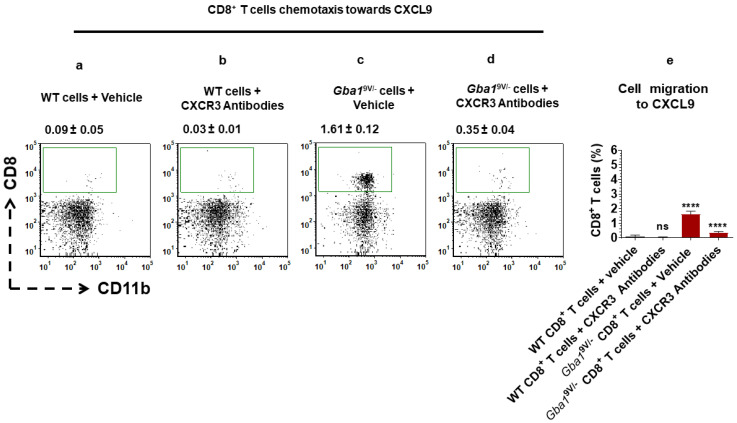
CXCR3 targeting alters CXCL9-mediated CD8^+^ T cells chemotaxis in *Gba1*^9V/−^ mice. CD8^+^ T cells purified from spleens of WT and *Gba1*^9V/−^ mice (*n* = 5/group) were allowed to migrate towards CXCL9 (16 nM) in the presence and absence of mouse anti-CXCR3 antibodies (10 μg/mL) at 37 °C and 5% CO_2_ for 45 min. Cells that had migrated through the filter and had attached to the lower side of the filter were collected and analyzed by FACS. Percentage of CD8^+^CD11b^−^ T cells are shown from the (**a**) vehicle (PBS) treated WT cells and their migration to CXCL9, (**b**) mouse anti-CXCR3 antibodies treated WT cells and their migration to CXCL9, (**c**) PBS treated *Gba1*^9V/−^ cells and their migration to CXCL9, and (**d**) mouse anti-CXCR3 antibodies treated *Gba1*^9V/−^ cells and their migration to CXCL9. (**e**) WT (black columns), *Gba1*^9V/−^ (maroon columns) and the values shown in the bar diagram are the mean ± SD. and group comparison was performed with ANOVA. Three independent experiments were conducted (**** *p* < 0.0001).

**Figure 6 ijms-22-12712-f006:**
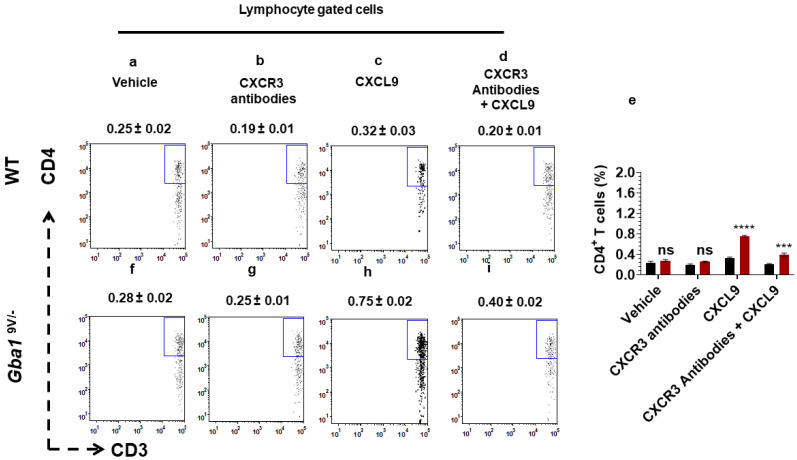
In vivo blocking of CXCR3 alters the CXCL9- mediated CD4^+^ T cells chemotaxis in *Gba1*^9V/−^ mice. WT and *Gba1*^9V/−^ mice were injected with intraperitoneal administration of CXCL9 and its vehicle as described in the method. In additional experiments, these mice were injected with intravenous injection of mouse anti-CXCR3 antibodies prior to intraperitoneal injection of CXCL9 or vehicle and the peritoneal cells were collected and analyzed by FACS. The dot plots and the corresponding bar diagrams represent the percentage of migrated CD3^+^CD4^+^ T cells in WT (**a**–**e**) and *Gba1*^9V/−^ mice (**e**–**i**). WT (black columns), *Gba1*^9V/−^ (maroon columns) and the values shown are the mean ± SD. and group comparison were performed with ANOVA. Three independent experiments were conducted (ns, not significant; ***, *p* < 0.001, ****, *p* < 0.0001).

**Figure 7 ijms-22-12712-f007:**
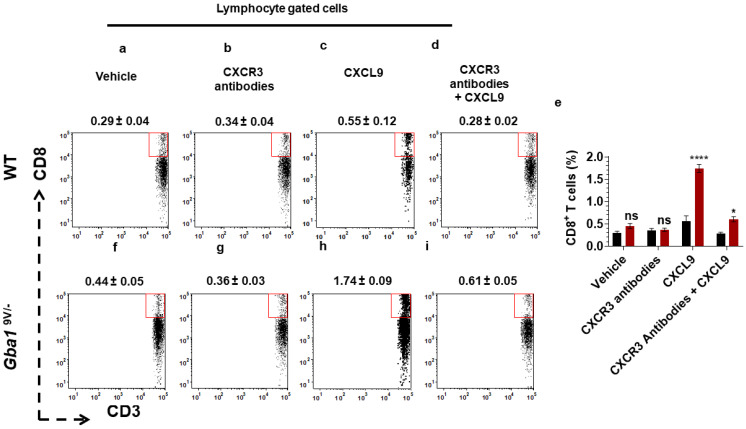
In vivo blocking of CXCR3 alters the CXCL9-mediated CD8^+^ T cells chemotaxis in *Gba1*^9V/−^ mice. WT and *Gba1*^9V/−^ mice were injected with intraperitoneal administration of CXCL9 and its vehicle as described in the method. In additional experiments, these mice were injected with intravenous injection of mouse anti-CXCR3 antibodies prior to intraperitoneal injection of CXCL9 or vehicle and the peritoneal cells were collected and analyzed by FACS. The dot plots and the corresponding bar diagrams represent the percentage of migrated CD3^+^CD8^+^ T cells in WT (**a**–**e**) and *Gba1*^9V/−^ mice (**e**–**i**). WT (black columns), *Gba1*^9V/−^ (maroon columns), and the values shown are the mean ± SD. and group comparison were performed with ANOVA. Three independent experiments were conducted (ns, not significant; * *p* < 0.05; **** *p* < 0.0001).

## Data Availability

Not applicable.

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
