# Peer review of "C-X-C Motif Chemokine Ligand 9 and Its CXCR3 Receptor Are the Salt and Pepper for T Cells Trafficking in a Mouse Model of Gaucher Disease"

_ijms, 2021, doi:10.3390/ijms222312712_

Round 1

Reviewer 1 Report

Major issues:

  1. CXCR3 has at least 3 cognate ligands, CXCL9, 10 and 11. Why was CXCL9 only studied for the disease model?
  2. The authors examined the expression of CXCR3 on T cells derived from the lungs. For chemotaxis purpose, the T cells from the blood should be tested instead of those from the tissue, because the receptor level is important before not after migrating out of blood vessels. Besides, when cells are exposed to high concentrations of ligands, the receptor is usually internalized and therefore lower after migration.
  3. Using CXCR3 antibody to test whether CXCL9 and CXCR3 are responsible for the T cell migration in the disease model does not provide any new finding. It is known that CXCL9 is chemokine for CXCR3 and that as a chemokine, CXCL9 should induce chemotaxis of CXCR3-expressing cells. The right assays are testing the effects of CXCR3 antibody treatment in the disease model without injection of CXCL9 or effects of CXCR3 antagonist on the T cell accumulation in the disease area in the disease model without additional injection of CXCL9.
  4. Line 124: Results expressed as the mean value of triplicate samples. For statistic analysis, results should be collected from at least three independent experiments.
  5. Figure 1: 1d, in Gba9V/- cells, MFI >2000, but in 1c, most cells have FI <1000? Therefore, the bar graph results do not agree with the original flow graph. The same is in 1g and 1h? The y axis in the histograms is cell numbers, not MFI.
  6. Figure 2: has the same problem as Fig 1.
  7. Figure 3: 1). 1b and 1d should use curves instead of columns; 2). The range of CXCL9 concentrations should be much wider, from very low to very high concentrations to see the whole dose-dependent responses; 3). The comparison with the 0 nM within one mouse group is meaningless, should be compared between two mouse groups.
  8. Figure 4: In 4e, column 3 is more than 10 times higher than column 1 while the difference is only about 2 in figure 3?
  9. Figure 5: 1). On the top, it says … chemotaxis toward … buffer, but actually CXCL9; 2). The comparison between what is not clearly stated.
  10. Suppl figure 3 and text lines 259-268 indicate that in WT mice, CXCL9 induces T cell chemotaxis, which can be blocked by antibody against CXCR3, contradictory to the results in Figures 4 and 5.

Minor issues: 

   There are a lot of typo or grammar errors. Just list a few as below:

  1. Line 62-65: The genetic deficiency or pharmaceutical blockade of complement 5a receptor (C5aR) resulted in reduction of activated subsets if CD4+ T cells and pro-inflammatory cytokines in co-cultured cells, 
  2. Line 106: Lungs tissues from WT and Gba1 9V/-mice were removed aseptically.
  3. Line 111: Mɸs DCs, CD4+ T lymphocytes,
  4. Line 163: (106/cells/200μl of complete RPMI media) for 48h

  5. Line 170: Values one asterisks, p<0.05); two asterisks, p<0.01); three asterisks, p<0.001); four asterisks, p<0.0001) were considered statistically significant.

Reviewer 2 Report

General comments:

  1. The title and abstract erroneously give the impression that a role for endogenous CXCL9 was uncovered in this study. Please consider revising it or include experiments with CXCL9-targeting antibodies to provide insight into the role of endogenous CXCL9 in this mouse model.
  2. The key words provided do not accurately capture the scope of this article.
  3. The introduction section would benefit from a more accurate description of GD (e.g. clinical symptoms).
  4. Please provide scientific rationale for focusing on CXCL9 while the other CXCR3 ligands (i.e. CXCL10 and CXCL11) are more powerful T and NK cell chemoattractants.
  5. Hampering the translational potential of this research is the fact that human CXCL9-11, in contrast to murine CXCL9 and CXCL11, are inactivated rapidly in the presence of physiological concentrations of dipeptidyl peptidase IV/CD26. Please address this issue in the discussion section.
  6. Validation experiments are required to confirm successful genetic ablation of Gba1.
  7. Are there any effects of genetic ablation of Gba1 on phenotype, (immune cell) development and/or number of circulating immune cells (in particular T cells, macrophages and DCs)?
  8. Are there any direct effects of genetic ablation of Gba1 on the function and adhesive properties of T cells and/or their capacity to migrate toward a source of chemoattractants (in particular CXCR3 ligands)? This could also affect attachment to chemotaxis membranes and affect the amount of cells detected on the filters or in the compartment below the filters (this fraction is not counted !) in WT versus Gba1 9v/- mice.
  9. Please provide rationale for studying lung versus spleen cells in the different experiments.
  10. Experiments with primary T lymphocytes from patients with GD would strengthen the manuscript.
  11. In general, the language quality is poor and unacceptable. Please correct grammatical errors and typos throughout the text.

Specific comments:

  1. Line 38: please indicate the prevalence of GD. Note that the description of GD as a ‘common disease’ is in conflict with the first key term provided in line 16 of the manuscript.
  2. Line 43: the descriptor ‘poorly defined immunological cell populations’ seems inappropriate.
  3. Line 71: CXCR3 ligands do also attract and activate NK cells.
  4. Line 105: please include study number/reference number of the ethical committee.
  5. Line 107: did cell preparation procedures induce/affect T cell activation and/or viability?
  6. Line 117: did the authors include a reagent for visualization of living versus dead cells?
  7. Line 141: 8µm pores are large for lymphocyte chemotaxis experiments. Moreover, an incubation period of 45 minutes is unusually short for these cells. Please explain!
  8. Line 12: 10 µg is an amount, not a concentration. What was the concentration?
  9. Line 153: please include specifications of the CXCR3 antibody used in ex vivo experiments.
  10. Line 156: What do you mean with “using a vertical glass on the top of 50 ml falcon tube”?
  11. Line 181: did the authors examine the putative effects of Gba1 deficiency on the relative numbers/percentages of macrophages and/or DCs? Are there any effects of genetic ablation of Gba1 on the production of CXCL10 and CXCL11 by these cells?
  12. Line 187: figure legend is incomplete. What about the right plots in panels d and h, are these ELISA data? Were there any inducers used to provoke CXCL9 production (probably IFN-É£)?
  13. 1: neutrophils may also co-express CD11b and CD11c. Please include FSC/SSC plots and complete gating strategy in main file or as a supplementary figure.
  14. 1 c and g: Is this intracellular staining? CXCL9 is a secreted protein! Or do you detect CXCL9 sticking to the membrane of the cells?
  15. 1 panels d and h (right panels): are the concentration detected by flow cytometry or by ELISA? If by ELISA, how were cells kept in culture (duration, medium,…)?
  16. Line 208: please clearly indicate that these are ex vivo experiments and consider revising the title of this paragraph.
  17. 2: it looks like the same figure is erroneously included in panels d and h. Please correct if necessary.
  18. 3 title: these are cells “from” mice, NOT in mice!
  19. 3a,c: remarkable color differences exist between pictures shown in panels a and c. Please consider revising it. In addition, are these cytospins (as stated in the legend)? They appear as stained cells on chemotaxis membranes?
  20. 3b,d: were any significant differences obtained between results from WT and KO mice?
  21. 4: the authors should consider the use of different colors for WT and KO mice in order to make labels on the x-axis more concise.
  22. 4e: no migration is observed if WT cells are exposed to 16µM CXCL9. These findings are not in line with data shown in Fig. 3b. Please clarify. A comparable lack of migration to CXCL9 is shown in Fig. 5e for WT cells without antibody treatment. Is this correct?
  23. Was CXCL9 used for the experiments in Suppl. Fig. 1 and Suppl. Fig. 2? This is stated in the legend and probably wrong! It is also indicated on top of Suppl. Fig. 2!
  24. Number Suppl. Fig S3 as “Figure S3” not “Figure 3”
  25. Correct in the legend to Fig. S3 the part corresponding to the right bars in panels e and j: the migration is not to antibodies but to CXCL9 after i.v. treatment with antibodies to mouse CXCR3!
  26. 6-7: percentages are extremely low and the biological relevance of these data is unclear. Statistical analyses is needed between CXCL9 treated mice and mice treated with CXCL9 and i.v. anti CXCR3 antibodies. Is this difference significant. If not, then only a trend is shown and the titles of the figure 6 and 7 would be an overstatement. Indicate in the figure legend how much CXCL9 or antibody was injected. Microscopic evaluation of peritoneal cell infiltrates would strengthen the conclusions. The authors should consider studying the CXCL9-induced migratory behavior of (fluorescently) labeled T cells in the presence and absence of CXCR3 antibodies as a proof of concept experiment.
  27. Line 316: authors state that macrophages and DCs are the predominant CXCL9-producing cells. However, other cells such as endothelial cells and fibroblasts may also produce this chemokine. Since macrophages and DCs are the only cells included in the present study, no conclusions can be made regarding major sources of CXCL9 in Gba1 deficient mice. Please consider revising the statement made in line 316.

Reviewer 3 Report

see attached file

Round 2

Reviewer 1 Report

  1. To make a solid conclusion that CXCL9 and its CXCR3 receptor account for the enhanced T cells trafficking in a mouse model of the  Gaucher disease, it is important to test T cell migration in the model without injecting exogenous CXCL9 using either CXCL9 or CXCR3 antibody or antagonist. The authors respond with "testing that in the future study". I am comfortable with that.
  2. In Fig. 1 and 2, the histograms and bar graphs are still confusing. I don't understand how they got the numbers in the bar graphs. I know the MFI is delta MFI (the specific-isotype), but the MFIs are based on flow data, histograms. Besides, the y axis of histograms should be relative cell numbers not MFI.

Reviewer 2 Report

  1. Concerning the authors' answer related to the use of lung or blood-derived cells (comment of reviewer 1). A Boyden chamber chemotaxis experiment (48 wells with 50 µl/well) can be performed with a few million cells. You don't need more than 25 million cells. However, I agree that you still would need to sacrifice multiple mice. I advise to indicate the concerns of downregulation of chemokine receptor expression once cells migrated to tissues (in this case the lung) as a limitation of the study.
  2. The manuscript still needs significant grammar and spelling corrections before publication. (too many to list individually!)

Unfortunately I failed to open the supplementary files and could not evaluate them.
